# LCS: Learning Compressible Subspaces for Adaptive Network Compression at Inference Time

## Abstract

When deploying deep learning models to a device, it is traditionally assumed that available computational resources (compute, memory, and power) remain static. However, real-world computing systems do not always provide stable resource guarantees. Computational resources need to be conserved when load from other processes is high or battery power is low. Inspired by recent works on neural network subspaces, we propose a method for training a *compressible subspace* of neural networks that contains a fine-grained spectrum of models that range from highly efficient to highly accurate. Our models require no retraining, thus our subspace of models can be deployed entirely on-device to allow adaptive network compression at inference time. We present results for achieving arbitrarily fine-grained accuracy-efficiency trade-offs at inference time for structured and unstructured sparsity. We achieve accuracies on-par with standard models when testing our uncompressed models, and maintain high accuracy for sparsity rates above 90% when testing our compressed models. We also demonstrate that our algorithm extends to quantization at variable bit widths, achieving accuracy on par with individually trained networks.

## 1 Introduction

Deep neural networks are deployed to a variety of computing platforms, including phones, tablets, and watches (Dhar et al., 2019). Networks are generally designed to consume a fixed budget of resources, but the compute resources available on a device can vary while the device is in use. Computational burden from other processes, as well as battery life, may influence the availability of resources to a neural network. Adaptively adjusting inference-time load is beyond the capabilities of traditional neural networks, which are designed with a fixed architecture and a fixed resource usage.

A simple approach to the problem of providing an accuracy-efficiency trade-off is to train multiple networks of different sizes. In this approach, multiple networks are stored on the device and loaded into memory when needed. There is a breadth of research in the design of efficient architectures that can be trained with different capacities, then deployed on a device (Howard et al., 2017; Sandler et al., 2018; Howard et al., 2019). However, there are a few drawbacks with using multiple networks to provide an accuracy-efficiency trade-off: (1) it requires training and deploying multiple networks (which induces training-time computational burden and on-device storage burden), (2) it requires all compression levels to be specified before deployment, and (3) it requires new networks to be loaded into memory when changing the compression level, which prohibits real-time adaptive compression.

Previous methods address the first issue in the setting of structured sparsity by training a single network conditioned to perform well when varying numbers of channels are pruned (Yu et al., 2018; Yu & Huang, 2019). However, these methods require BatchNorm (Ioffe & Szegedy, 2015) statistics to be recalibrated for every accuracy-efficiency configuration before deployment. This requires users to know every compression level in advance and requires storage of extra parameters (which can be problematic for low-compute devices). Another method trains a larger network that can be queried for efficient subnetworks (Cai et al., 2020), but requires users to reload network weights, and to know every compression level before deployment.

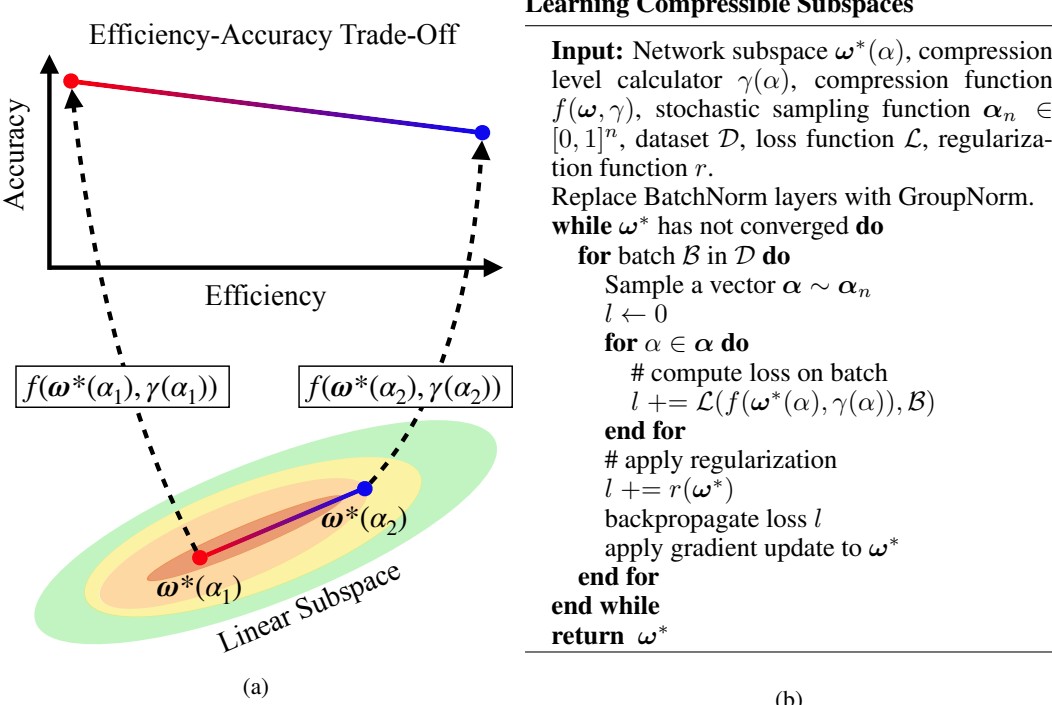

**Learning Compressible Subspaces**

**Input:** Network subspace $\omega^*(\alpha)$, compression level calculator $\gamma(\alpha)$, compression function $f(\omega, \gamma)$, stochastic sampling function $\alpha_n \in [0,1]^n$, dataset $\mathcal{D}$, loss function $\mathcal{L}$, regularization function $r$.
Replace BatchNorm layers with GroupNorm.
**while** $\omega^*$ has not converged **do**
    **for** batch $\mathcal{B}$ in $\mathcal{D}$ **do**
        Sample a vector $\alpha \sim \alpha_n$
        $l \leftarrow 0$
        **for** $\alpha \in \alpha$ **do**
            # compute loss on batch
            $l \mathrel{+}= \mathcal{L}(f(\omega^*(\alpha), \gamma(\alpha)), \mathcal{B})$
        **end for**
        # apply regularization
        $l \mathrel{+}= r(\omega^*)$
        backpropagate loss $l$
        apply gradient update to $\omega^*$
    **end for**
**end while**
**return** $\omega^*$

(a)

(b)

Figure 1: (a) Depiction of our method for learning a linear subspace of networks $\omega^*$ parameterized by $\alpha \in [\alpha_1, \alpha_2]$. When compressing with compression function $f$ and compression level $\gamma$, we obtain a spectrum of networks which demonstrate an efficiency-accuracy trade-off. (b) Our algorithm.

We address the issue of deploying a model that can be compressed in real-time without specifying the compression levels beforehand. Inspired by Wortsman et al. (2021), we formulate this problem as computing an adaptively compressible network *subspace* (Figure 1a). We train a linear subspace using two sets of weights, conditioning them so that a convex combination of them produces an accurate network. We expand on Wortsman et al. (2021) by (1) specializing our linear subspace to contain an *arbitrarily fine-grained efficiency-accuracy trade-off* (meaning, any compression level within predefined limits can be chosen), and (2) replacing BatchNorm (Ioffe & Szegedy, 2015) with GroupNorm (Wu & He, 2018) to avoid BatchNorm recalibration.

**Contributions:** Our contributions are as follows. (1) We introduce our method, Learning Compressible Subspaces (LCS), for training models that can be compressed with arbitrarily fine-grained compression levels in real-time after deployment. To our knowledge this has not been done before. (2) We demonstrate a new use for neural network subspaces, specializing at one end for high-accuracy and at the other end for high-efficiency. (3) We provide an empirical evaluation of our method using structured sparsity, unstructured sparsity, and quantization. (4) We open source our code at `https://withheld.for.review`.

## 2 RELATED WORK

**Architectures Demonstrating Efficiency-Accuracy Trade-Offs:** Several recent network architectures include a hyperparameter to control the number of filters in each layer (Howard et al., 2017; Sandler et al., 2018; Howard et al., 2019) or the number of blocks in the network (Tan & Le, 2019). In Once For All (Cai et al., 2020), the need for individually training networks of different sizes is circumvented. Instead, a single large network is trained, then queried for efficient subnetworks.

We differ from these methods in two ways. First, we compress on-device without specifying the compression levels before deployment. Previous works require training separate networks (or in

Table 1: Our method with a linear subspace (LCS+L) and a point subspace (LCS+P) compared to LEC (Liu et al., 2017), NS (Yu et al., 2018), and US (Yu & Huang, 2019). Note that "Arbitrary Compression Levels" refers to arbitrarily fine-grained post-deployment compression. $|\boldsymbol{\omega}|$ denotes the number of network parameters, $|b|$ denotes the number of BatchNorm parameters, and $n$ denotes the number of compression levels for models that don't support arbitrarily fine-grained compression.

| | LCS+P | LCS+L | LEC | NS | US |
|---|---|---|---|---|---|
| No Retraining | ✔ | ✔ | ✘ | ✔ | ✔ |
| No Norm Recalibration | ✔ | ✔ | ✘ | ✘ | ✘ |
| Arbitrary Compression Levels | ✔ | ✔ | ✘ | ✘ | ✘ |
| Stored Parameters | $|\boldsymbol{\omega}|$ | $2|\boldsymbol{\omega}|$ | $n|\boldsymbol{\omega}|$ | $|\boldsymbol{\omega}| + n|b|$ | $|\boldsymbol{\omega}| + n|b|$ |

the case of Once For All, querying a larger model for a compressed network) before deployment. Second, we don't require deploying a set of weights for every compression level.

A method for post-training quantization to variable bit widths appears in Shkolnik et al. (2020), but it relies on calibration data and cannot be run on low-compute edge devices. Thus, compression levels must be specified before deployment. They also present a method for quantization-aware training to variable bit widths, but most of their results keep activation bit widths fixed, whereas we vary it.

**Adaptive Compression:** Recent works train a single neural network which can be adaptively configured at inference time to execute at different compression levels. These methods are the closest to our work. In Learning Efficient Convolutions (LEC) (Liu et al., 2017), the authors train a single network, then fine-tune it at different structured sparsity rates (though they also present results without fine-tuning, which is closer to our use case). Other methods train a single set of weights conditioned to perform well when channels are pruned, but require recalibration (or preemptive storage) of BatchNorm (Ioffe & Szegedy, 2015) statistics at each sparsity level. These methods include Network Slimming (NS) (Yu et al., 2018) and Universal Slimming (US) (Yu & Huang, 2019). Similar methods train a single network to perform well at various levels of quantization by storing extra copies of BatchNorm statistics (Guerra et al., 2020).

We differ from these methods by avoiding the need to recalibrate or store BatchNorm statistics (Section 3.3) and by allowing for fine-grained selection of compression levels at inference time, neither of which have been done before (Table 1). Additionally, our method is broadly applicable to a variety of compression methods, whereas previous works focus on a single compression method.

**Other Post-Training Compression Methods:** Other works have investigated post-training compression. In Nagel et al. (2019), a method is presented for compressing 32 bit models to 8 bits, though it has not been evaluated in the low-bit regime. It involves running an equalization step and assuming a Conv-BatchNorm-ReLU network structure. A related post-training compression method is shown in Horton et al. (2020), which shows results for both quantization and sparsity. However, their sparsity method requires a lightweight training phase. We differ from these methods in providing real-time post-deployment compression for both sparsity and quantization without making assumptions about the network structure.

**Sparse Networks:** Sparsity is a well-studied approach to reduce neural network compute. Recent works have achieved state-of-the-art accuracy using simple magnitude-based pruning (Han et al., 2015; Zhu & Gupta, 2017; See et al., 2016; Narang et al., 2017). When experimenting with structured sparsity, we simply remove filters from the network, as described in Yu et al. (2018). When investigating unstructured sparsity, we use TopK pruning (Zhu & Gupta, 2017).

**Quantized Networks:** Network quantization represents the weights and/or activations of neural networks with lower-bit integral representations, rather than 32-bit floating point. This reduces model size and memory bandwidth and can accelerate inference. Our work follows the affine quantization scheme presented in Jacob et al. (2018), in which a scale and offset are applied to an integer tensor to approximate the original floating-point tensor. However, our method is general enough to utilize any other quantization scheme.

**Neural Network Subspaces:** The idea of learning a neural network subspace is introduced in Wortsman et al. (2021) (though another formulation was introduced concurrently in Benton et al. (2021)).

Multiple sets of network weights are treated as the corners of a simplex, and an optimization procedure updates these corners to find a region in weight space in which points inside the simplex correspond to accurate networks. We note that Wortsman et al. (2021) recalibrate BatchNorm statistics before evaluation, which we avoid.

## 3 COMPRESSIBLE SUBSPACES

### 3.1 COMPRESSIBLE LINES

In our method, we train a neural network subspace (Wortsman et al., 2021) that contains a spectrum of networks that each have a different efficiency-accuracy trade-off. In Wortsman et al. (2021), an architecture is chosen, and its collection of weights is denoted by $\omega$. A linear subspace is parameterized by two sets of network weights, $\omega_1$ and $\omega_2$. At training time, a network is sampled from the line defined by this pair of weights, $\omega^*(\alpha) = \alpha\omega_1 + (1-\alpha)\omega_2$, where $\alpha \in [0, 1]$. The forward and backward pass of gradient descent are computed using $\omega^*(\alpha)$, and gradient updates are backpropagated to $\omega_1$ and $\omega_2$. A cosine distance regularization term is used to ensure that $\omega_1$ and $\omega_2$ don't converge to a single point.

We propose training a linear subspace of networks in a similar manner, but we also bias the subspace to contain high-accuracy solutions at one end and high-efficiency solutions at the other end. To accomplish this, we introduce a parameter $\gamma$ used to control the compression level and a compression function $f(\omega, \gamma)$ used to compress a model's weights $\omega$ using a compression level $\gamma$. The parameter $\gamma$ induces different compression levels at different positions in the linear subspace, and therefore can be considered a function $\gamma \equiv \gamma(\alpha)$. When training, we first sample a network from the subspace, then compress it, obtaining a network with weights $f(\omega^*(\alpha), \gamma(\alpha))$. We then perform a standard forward and backward pass of gradient descent with it, backpropagating gradients to $\omega_1$ and $\omega_2$.

In summary, our training method adjusts the standard subspace training method to include the application of a compression function $f$ at a compression level $\gamma(\alpha)$. Note that we apply the compression after sampling the model $\omega^*(\alpha)$ from the subspace, since we want the sampled model to be compressed (rather than $\omega_1$ and $\omega_2$) for efficient inference. We experiment with formulating our compression function as structured sparsity, unstructured sparsity, and quantization (Section 4). In all cases, the cost of computing $f$ is small compared to the cost of a network forward pass (Section A.1).

### 3.2 COMPRESSIBLE POINTS

In Section 3.1, we discussed formulating our subspace as a line connected by two endpoints in weight space. This formulation requires additional storage resources to deploy the subspace (Table 1), since an extra copy of network weights is stored. For some cost-efficient computing devices, this storage may be significant. To eliminate this need, we propose training a degenerate subspace with a single point in weight-space (rather than two endpoints). We still use $\alpha \in [0, 1]$ to control our compression ratio, but our subspace is parameterized by a single set of weights, $\omega^*(\alpha) = \omega$. The compressed weights are now expressed as $f(\omega^*(\alpha), \gamma(\alpha)) \equiv f(\omega, \gamma(\alpha))$. This corresponds to applying varying levels of compression during each forward pass. The cosine distance regularization term included in the linear subspace is no longer needed because only one set of weights is trained.

This method still produces a subspace of models in the sense that, for each value of $\alpha$, we obtain a different compressed network $f(\omega, \gamma(\alpha))$. However, we no longer use different endpoints of a linear subspace to specialize one end of the subspace for accuracy and the other for efficiency. Instead, we condition one set of network weights to tolerate varying levels of compression.

### 3.3 CIRCUMVENTING BATCHNORM RECALIBRATION

Previous works that train a compressible network require an additional training step to calibrate BatchNorm (Ioffe & Szegedy, 2015) statistics, as in Universal Slimming (Yu & Huang, 2019). Alternatively, they require storing separate BatchNorm statistics for each compression level, as in Slimmable Networks (Yu et al., 2018). This precludes both methods from evaluating at arbitrarily fine-grained compression levels after deployment (Table 1). We seek to eliminate the need for recalibration or storage.

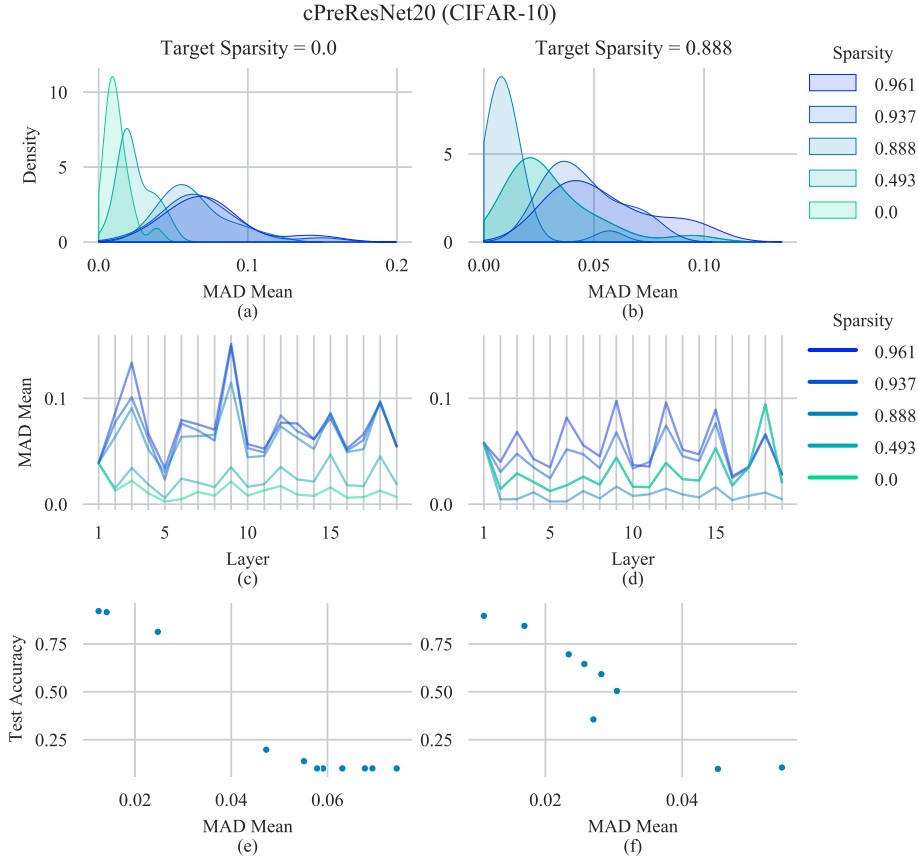

Figure 2: Analysis of observed batch-wise means $\hat{\mu}$ and stored BatchNorm means $\mu$ during testing for models trained with TopK unstructured sparsity. The models are trained with different target sparsities and evaluated with various inference-time sparsities. (a)-(b): The distribution of $|\mu - \hat{\mu}|$ across all layers. (c)-(d): The average value of $|\mu - \hat{\mu}|$ for individual layers. (e)-(f): The correlation between the average of $|\mu - \hat{\mu}|$ and test set error. Note that in (b) and (d), sparsities of $0$ and $0.493$ produce near-identical results, thus those curves are overlapping.

To understand the need for recalibration in previous works, recall that BatchNorm layers store the per-channel mean of the inputs $\mu$ and the per-channel variance of the inputs $\sigma^2$. The recalibration step is needed to correct $\mu$ and $\sigma^2$, which are corrupted when a network is adjusted. Adjustments that corrupt statistics include applying structured sparsity, unstructured sparsity, and quantization.

In Figure 2, we analyze the inaccuracies of BatchNorm statistics for two models trained with specific unstructured sparsity levels and tested with a variety of inference-time unstructured sparsity levels. We calculate the differences between stored BatchNorm means $\mu$ and the true mean of a batch $\hat{\mu}$ during the test epoch of a cPreResNet20 (He et al., 2016) model on CIFAR-10 (Krizhevsky, 2009). In Figure 2a and Figure 2b, we show the distribution of mean absolute differences (MADs) $|\mu - \hat{\mu}|$ across all layers of the models. Models have lower BatchNorm errors when evaluated near sparsity levels that match their training-time target sparsity. Applying mismatched levels of sparsity shifts the distribution of these errors away from $0$. In Figure 2c and Figure 2d, we show the average of $|\mu - \hat{\mu}|$ across the test set for each of the BatchNorm layers. Across layers, the lowest error is achieved when the level of sparsity matches training. In Figure 2e and Figure 2f, we show the average of $|\mu - \hat{\mu}|$ and the corresponding test set accuracy for various levels of inference-time sparsity. We find that the increased error in BatchNorm is correlated with decreased accuracy. See Figure 7 and Figure 8 in the Appendix for similar analyses with structured sparsity and quantization.

Thus, BatchNorm layers' stored statistics can become inaccurate during inference-time compression, which can lead to accuracy degradation. To circumvent the need for BatchNorm, we adjust our

networks to use GroupNorm (Wu & He, 2018). This computes an alternative normalization over $g$ groups of channels rather than across a batch. It doesn't require maintaining a running average of the mean and variance across batches of input, so there are no stored statistics that can be corrupted if the network changes. Formally, let $i$ be an index into a 4-dimensional image tensor, with components $i = [i_N, i_C, i_H, i_W]$ indexing the batch, channel, height, and width, respectively. The set $\mathbb{S}_i$ of indices used to compute the normalization of batch element $i$ is

$$\mathbb{S}_i = \{k | k_N = i_N, \lfloor k_C * g/c \rfloor = \lfloor i_C * g/c \rfloor\}. \tag{1}$$

Batch element $x_i$ is normalized by subtracting the mean of elements indexed by $\mathbb{S}_i$ and dividing by the standard deviation of elements indexed by $\mathbb{S}_i$. GroupNorm typically uses the hyperparameter $g = 32$, but it also includes InstanceNorm (Ulyanov et al., 2016) (in which $g = c$, where $c$ is the number of channels) as a special case. We use $g = c$ in structured sparsity experiments, since the number of channels is determined dynamically and is not always divisible by 32. We use $g = 32$ in all other experiments [1]. After applying this normalization, we also apply a learned per-channel affine transformation, as with BatchNorm.

## 3.4 BIASED SAMPLING

The endpoints of a learned linear subspace typically demonstrate slightly reduced accuracy compared to the center of the line, as observed in Wortsman et al. (2021). Inspired by this observation, we use biased sampling to choose network endpoints more frequently when training a linear subspace, to ensure that endpoints are accurate.

A similar algorithmic choice is made in Yu & Huang (2019), which proposes using a "sandwich method" for training. This method involves performing forward and backward passes at the maximum and minimum sparsity levels, as well as a few randomly chosen sparsity levels in between. After this round of forward and backward passes, the gradient update is applied.

To implement our sampling of $\alpha$, we use a stochastic function $\boldsymbol{\alpha}_n : \boldsymbol{\Omega} \to [0, 1]^n$, where $\boldsymbol{\Omega}$ represents the state of the stochastic function. For each batch, we sample $\boldsymbol{\alpha} \sim \boldsymbol{\alpha}_n$. We perform $n$ forward and backward passes using compressed networks $f(\boldsymbol{\omega}^*(\alpha_i), \gamma(\alpha_i))$ for $i \in \{1, ..., n\}$, where $\alpha_i$ is the $i^{th}$ element of $\boldsymbol{\alpha}$. The stochastic function used varies based on the compression function chosen. See Section 4.1 (structured sparsity), Section 4.2 (unstructured sparsity), and Section 4.3 (quantization) for more details. Our full method for training a compressible subspace is shown in Figure 1b.

## 4 EXPERIMENTS

We present results in the domains of structured sparsity, unstructured sparsity, and quantization. We train using Pytorch (Paszke et al., 2019) on Nvidia Tesla V100 GPUs. On CIFAR-10 (Krizhevsky, 2009), we experiment with the pre-activation version of ResNet20 (He et al., 2016) presented in the PyTorch version of the open-source code provided by Liu et al. (2017). We abbreviate it as "cPreResNet20" to denote that it's a pre-activation network designed for usage with CIFAR-10. We use a pre-activation version because our Learning Efficient Convolutions (LEC) (Liu et al., 2017) baseline requires it. We follow hyperparameter choices in Wortsman et al. (2021) for our methods and baselines. We warm up to our initial learning rate of 0.1 (or 0.025 in the case of quantization, for stability) over 5 epochs, then decay to 0 over 195 epochs using a cosine schedule. We use a batch size of 128 on a single GPU. We set the weight decay to $5 \times 10^{-4}$. We run 3 trials for each experiment and report the mean and standard deviation (which is often too small to be visible). Our baseline cPreResNet (with BatchNorm) achieves an accuracy of $91.69\%$ on CIFAR-10.

For ImageNet (Deng et al., 2009) experiments, we use ResNet18 (He et al., 2016) and VGG19 (Simonyan & Zisserman, 2014). We use the version of VGG19 provided by the same repository as our cPreResNet20 implementation. This implementation modifies VGG19 slightly by adding BatchNorm layers and removing the last two fully connected layers. Again, we follow hyperparameter settings in Wortsman et al. (2021) for our methods and baselines. We warm up to our initial learning rate of 0.1 (or 0.025 in the case of quantization, for stability) over 5 epochs, then decay to 0 over 85 epochs using a cosine schedule. We set the weight decay to $5 \times 10^{-5}$. We use a batch size of 128 for

---

[1] We set $g = c$ for the first few layers of cPreResNet20, because it has fewer than 32 channels.

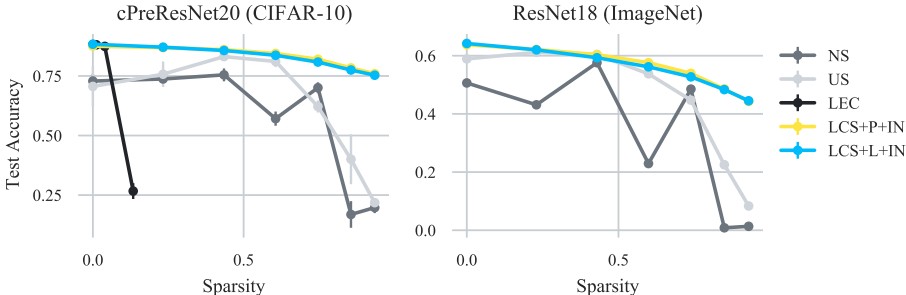

Figure 3: Our method for structured sparsity using a linear subspace (LCS+L+IN) and a point subspace (LCS+P+IN) compared to Universal Slimming (US) (Yu & Huang, 2019), Network Slimming (NS) (Yu et al., 2018), and Learning Efficient Convolutions (LEC) (Liu et al., 2017). LEC does not provide an open-source implementation of their method for ResNet18, so we omit it. We do not allow fine-tuning or recalibration.

ResNet18 with a single GPU, and a batch size of 256 for VGG19 with 4 GPUs. Unlike when training with CIFAR-10, we run a single trial for each experiment due to the increased computational expense of training on ImageNet. Our baseline ResNet18 (with BatchNorm) achieves an accuracy of 70.72% on ImageNet, and our VGG19 (with BatchNorm) achieves 62.21%.

## 4.1 STRUCTURED SPARSITY

For structured sparsity, our method follows the algorithm in Figure 1b. Our compression level calculator $\gamma(\alpha) : [0, 1] \rightarrow [0.25, 1.0]$ is the unique affine transformation over its domain and range. Our compression function $f(\omega, \gamma)$ prunes away a fraction $1 - \gamma$ of the input and output channels in each layer (except the input to the first layer and the output of the last layer). This is the same compression scheme used in Network Slimming (NS) (Yu et al., 2018) and Universal Slimming (US) (Yu & Huang, 2019). Our stochastic sampling function samples $\alpha$ as $[0.25, 1.0, U(0.25, 1), U(0.25, 1)]$, where $U(a, b)$ samples uniformly in the range $[a, b]$. This choice mirrors the "sandwich rule" used in US. As discussed in Section 3.3, we use a special case of GroupNorm (Wu & He, 2018) known as InstanceNorm (Ulyanov et al., 2016) since the number of channels in the network varies. See Section A.3 for additional details on our linear subspace setup.

When running baselines, we do not allow recalibration of BatchNorm (Ioffe & Szegedy, 2015) statistics (as used in US), or storage of a predefined set of BatchNorm statistics (as used in NS), since it would contradict our goal of adaptive compression to arbitrarily fine-grained sparsity levels at inference time. We instead use a single BatchNorm layer, using only the channels not removed by the structured pruning step. Similarly, we do not allow fine-tuning (normally used in LEC).

The main difference between our point method and US is that we use InstanceNorm instead of BatchNorm. The main difference between US and NS is that US includes the sandwich rule described above. Instead, NS samples at predefined sparsity levels $[0.25, 0.50, 0.75, 1]$ at each iteration.

We present results for structured pruning on ResNets in Figure 3 (see Section A.5 for VGG19 results and runtime characteristics). Our method produces a stronger efficiency-accuracy trade-off than competing methods. The poor performance of LEC (Liu et al., 2017) occurs because this method expects weights to be fine-tuned after sparsity is applied, which we do not allow. The dips and spikes shown for NS occur because this method trains at discrete width factors of $[0.25, 0.50, 0.75, 1]$, which roughly correspond to sparsities of $[0.9375, 0.75, 0.4325, 0]$. This method shows stronger accuracies at these sparsities, with the exception of $0.9375$, at which point, inaccuracies in BatchNorm statistics are more extreme (see Figure 7 in the Appendix). Similarly, US peaks in accuracy at moderate sparsity rates, but decreases in accuracy at high sparsity rates. We believe this strong decrease for higher sparsity rates occurs because the BatchNorm channels used in this case are also used at all other sparsity rates, thus the stored statistics don't accurately reflect the statistics of the highly-sparse network.

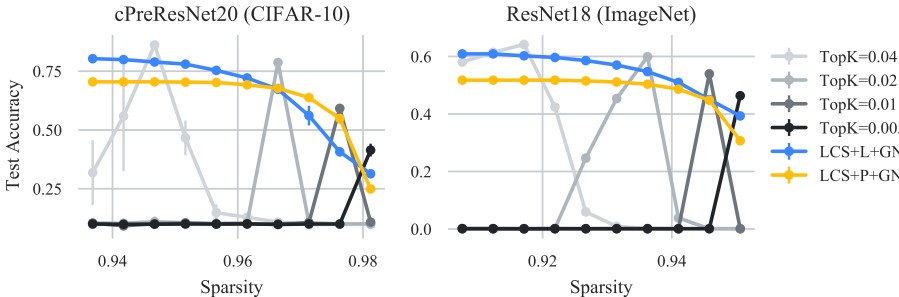

Figure 4: Our method for unstructured sparsity using a linear subspace (LCS+L+GN) and a point subspace (LCS+P+GN) compared to networks trained for a particular TopK target. The TopK target refers to the fraction of weights that remain unpruned during training.

Interestingly, our method with a linear subspace (LCS+L+IN) and our method with a point subspace (LCS+P+IN) achieve very similar performance. We believe that the extra weights in LCS+L+IN are not necessary because individual channels are able to specialize, since some of them are only used for larger subnetworks.

## 4.2 UNSTRUCTURED SPARSITY

For unstructured sparsity, our method follows the algorithm in Figure 1b. Our compression level calculator is $\gamma(\alpha) = 1 - \alpha$. Our compression function $f(\omega, \gamma)$ is TopK sparsity (Zhu & Gupta, 2017), which removes a fraction $\gamma$ of the weights with the smallest absolute value from each layer (except the input and output layer). Our stochastic sampling function samples one value for $\alpha$ in $[0.005, 0.05]$ (see Section A.4 for additional details regarding sampling and TopK's warmup phase). We choose this interval because our networks are shown to tolerate high sparsity levels, losing almost no accuracy even at 90% sparsity. See Section A.5 for experiments with broader ranges of sparsity, and for runtime characteristics of sparse models.

To our knowledge, fine-grained adaptive compression has not been explored for unstructured sparsity. Thus, we compare to standard networks (using BatchNorm) trained at specific sparsity levels and evaluated at a variety of sparsity levels. See Figure 4 for results. We find that our baselines always achieve highest accuracy at the sparsity level that matches their training-time TopK value, as expected. Accuracy quickly degrades at other sparsity levels. By contrast, our method is able to achieve high accuracy for a wide variety of sparsity levels. Interestingly, our line method usually achieves higher accuracy at lower sparsity rates in Figure 4, but this pattern does not hold over a wider sparsity range (Section A.5). Further investigation into this phenomena is future work.

## 4.3 QUANTIZATION

For quantization, our method follows Algorthm 1b. Our compression level calculator is $\gamma(\alpha) = 2 + 6\alpha$. Our compression function $f(\omega, \gamma)$ is affine quantization as described in (Jacob et al., 2018). Our stochastic sampling function samples one value for $\alpha$ uniformly over the set $\{1/6, 2/6, ..., 6/6\}$. This corresponds to training with bit widths 3 through 8 (we avoid lower bit widths to circumvent training instabilities we encountered in baselines). We train without quantizing the activations for the first 80% of training, then add activation quantization for the remainder of training. Weights are quantized throughout training.

We present results for our method in Figure 5, comparing to models trained at a fixed bit width and evaluated at a variety of bit widths. See Section A.5 for ResNet18 results, and for memory usage characteristics of models. Generally, baselines achieve high accuracy at the bit width at which they were trained, and reduced accuracy at other bit widths. Our method using a linear subspace (LCS+L+GN) achieves high accuracy at all bit widths, matching or exceeding accuracies of individual networks trained for target bit widths. In the case of VGG19, we found that our accuracy

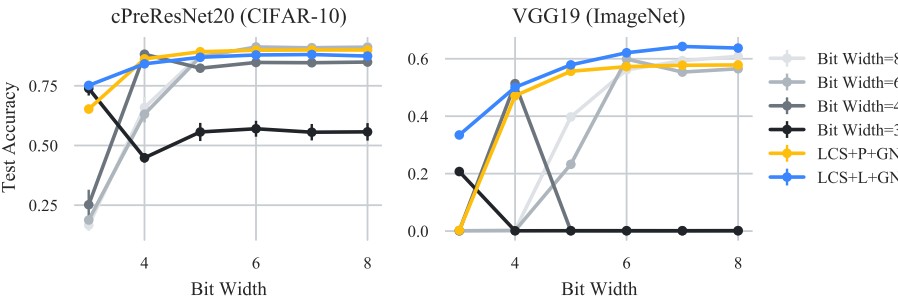

Figure 5: Our method for quantization using a linear subspace (LCS+L+GN) and a point subspace (LCS+P+GN) compared to networks trained for a particular bit width target.

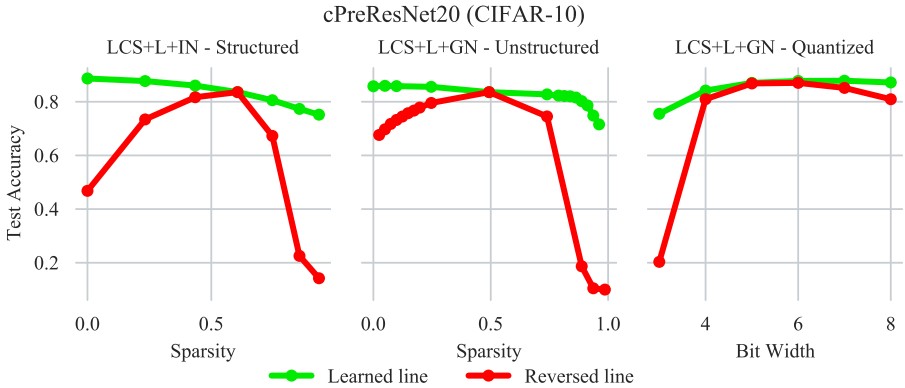

Figure 6: Standard evaluation of a linear subspace with network $f(\boldsymbol{\omega}^*(\alpha), \gamma(\alpha))$ (Learned line), and evaluation when evaluating with reversed compression levels, $f(\boldsymbol{\omega}^*(\alpha), \gamma(1-\alpha))$ (Reversed line).

even exceeded the baselines. We believe part of the increase is due to GroupNorm demonstrating improved results on this network compared to BatchNorm (which does not happen with ResNets, as reported in Wu & He (2018)).

### 4.4 FURTHER ANALYSIS

In Figure 6, we provide additional experimental evidence that our linear subspace method (LCS+L) trains a subspace specialized for high-accuracy at one end and high-efficiency at the other end. We plot the validation accuracy along our subspace, as well as the validation accuracy along our subspace when compressing with $\tilde{f}(\boldsymbol{\omega}^*(\alpha), \gamma(\alpha)) \equiv f(\boldsymbol{\omega}^*(\alpha), \gamma(1-\alpha))$. In other words, the weights that were trained for low compression levels are evaluated with high compression levels, and vice versa. We see that this leads to a large drop in accuracy, confirming that our method has conditioned one side of the line to achieve high accuracy at high sparsities, and the other side of the line to achieve high accuracy at low sparsities.

## 5 CONCLUSION

We present a method for learning a compressible subspace of neural networks. Our subspace can be compressed in real-time without retraining and without specifying the compression levels before deployment. We can produce a variety of models with a fine-grained efficiency-accuracy trade-off. We demonstrate that our generic algorithm can be applied to structured sparsity, unstructured sparsity, and quantization.

## 6 REPRODUCIBILITY STATEMENT

We open source our code to reproduce our results (including baselines) at `https://withheld.for.review`.

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

## A    APPENDIX

### A.1    COMPUTATIONAL COST OF COMPRESSION

We justify our claim that our network compression method is real-time. For a typical convolutional layer without downsampling, the computational cost is $\mathcal{O}(d_1 d_2 H_1 W_1 k_h k_w)$, where $d_1$ is the number of input channels, $d_2$ is the number of output channels, $H_1$ is the input tensor height, $H_2$ is the input

tensor width, $k_h$ is the kernel height, and $k_w$ is the kernel width. For our line method, we must calculate $\boldsymbol{\omega}^*(\alpha)$, which takes $\mathcal{O}(d_1 d_2 k_h k_w)$ for each convolution, which is negligible compared to the convolution. For our point method, calculating $\boldsymbol{\omega}^*(\alpha)$ requires no computation.

We must also calculate the compressed network, $f(\boldsymbol{\omega}^*(\alpha), \gamma(\alpha))$. Our structured sparsity method (described in Section 4.1) takes $\mathcal{O}(1)$ for each convolution, since we only need to mark each layer with the number of convolutional filters to ignore. Our unstructured sparsity method (TopK, described in Section 4.2) takes $\mathcal{O}(d_1 d_2 k_h k_w)$ for each convolution (to calculate the threshold, then discard parameters below the threshold). Our quantization method (Jacob et al., 2018) takes $\mathcal{O}(d_1 d_2 k_h k_w)$ for each convolution (to calculate the affine transform parameters and apply them). In all cases, the complexity of computing $f(\boldsymbol{\omega}^*(\alpha), \gamma(\alpha))$ is lower than the cost of the convolutional forward passes, meaning our method can be considered real-time.

## A.2 FURTHER BATCHNORM ANALYSIS

In Figure 2, we analyzed the inaccuracies of BatchNorm statistics for models trained with unstructured sparsity. We show a similar analysis for the case of Universal Slimming (US) (Yu & Huang, 2019) and Network Slimming (NS) (Yu et al., 2018) in Figure 7. We show the case of quantization in Figure 8.

## A.3 STRUCTURED SPARSITY DETAILS

When training our method with a line in the structured sparsity setting, we do not use two sets of weights (e.g. $\boldsymbol{\omega}_1$ and $\boldsymbol{\omega}_2$, Section 3.1) for convolutional filters. Instead, we only use two sets of weights for affine transforms in GroupNorm (Wu & He, 2018) layers. For the convolutional filters, we instead use a single set of weights, similar to our point formulation (and similar to US (Yu & Huang, 2019) and NS (Yu et al., 2018)). By contrast, we use two sets of weights for convolutional filters as well as for affine transforms when experimenting with unstructured sparsity (Section 4.2) and quantization (Section 4.3).

The reason for only using one set of convolutional filters in the structured sparsity setting is that the filters themselves are able to specialize, even without an extra copy of network weights. Some filters are only used in larger networks, so they can learn to identify different signals than the filters used in all subnetworks. Note that this filter specialization argument does not apply to our unstructured or quantized settings.

In preliminary experiments, we found that using a single set of weights for convolutions in our structured sparsity experiments gave a slight improvement over using two sets of weights (roughly 2% for cPreResNet20 (He et al., 2016) on CIFAR-10 (Krizhevsky, 2009)). We hypothesize that this slight difference may be attributed to the subspace training framework's regularization term (which increases cosine distance between $\boldsymbol{\omega}_1$ and $\boldsymbol{\omega}_2$), which discourages identical filters from being used in both $\boldsymbol{\omega}_1$ and $\boldsymbol{\omega}_2$ even if they are optimal.

## A.4 UNSTRUCTURED SPARSITY DETAILS

It is typical to include a warmup phase when training models with TopK sparsity (Zhu & Gupta, 2017). In our baselines in Section 4.2, we increase the sparsity level from 0% to its final value over the first 80% of training epochs. For our method, sparsity values fall within a range, so there is no single target sparsity value to warm up to.

For our point method (LCS+P), we simply train for the first 80% of training with the lowest sparsity value in our sparsity range. We finish training by sampling uniformly between the lowest and highest sparsity levels.

For our line method (LCS+L), our choice of sparsity level is tied to our choice of weight-space parameters through $\alpha$. We implement our warmup by simply adjusting $\gamma(\alpha)$ to apply less sparsity early in training, warming up to our final sparsity rates over the first 80% of training.

In detail, let $\alpha_{\min}$ and $\alpha_{\max}$ correspond to our minimum and maximum alpha values (for example, in Section 4.2, $\alpha_{\min} = 0.005$, and $\alpha_{\max} = 0.05$). As motivated in Section 3.4, we bias sampling of $\boldsymbol{\alpha}$ towards the endpoints of our line. We set $\boldsymbol{\alpha} = [\alpha_{\min}]$ with 25% probability, $\boldsymbol{\alpha} = [\alpha_{\max}]$

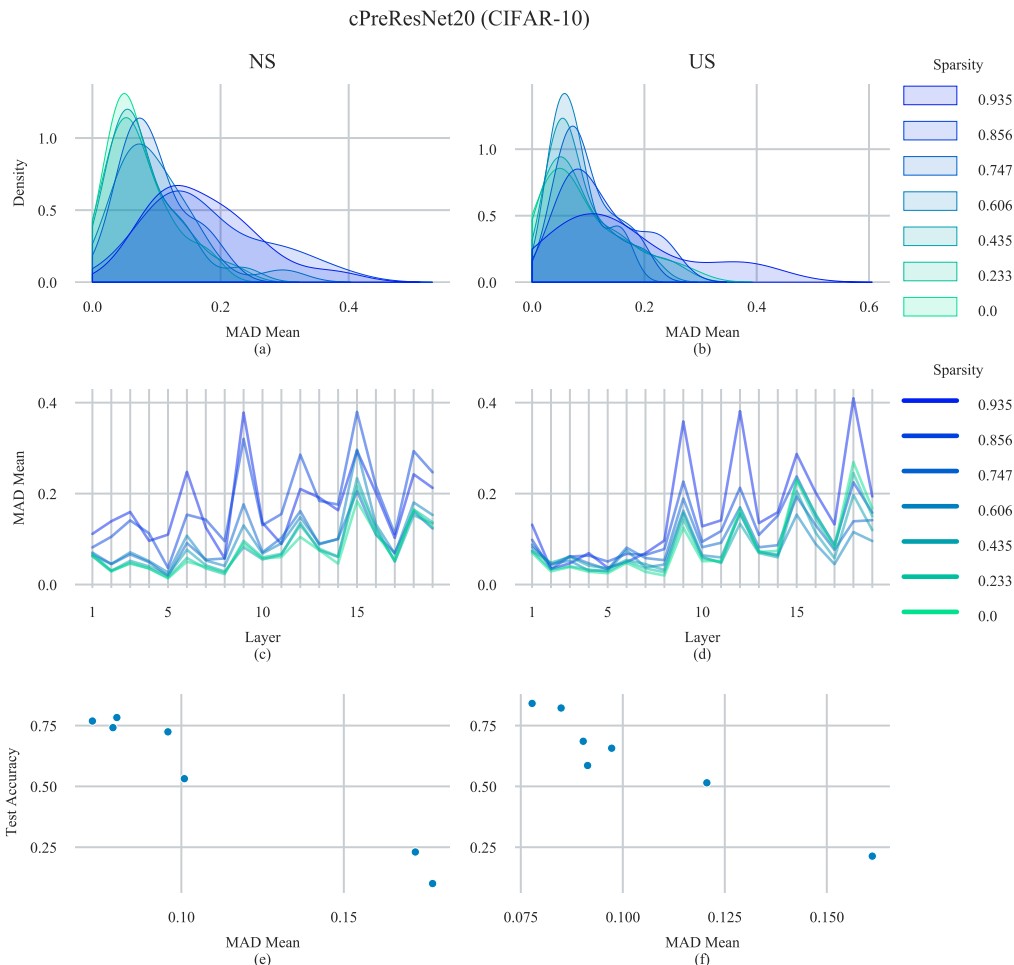

Figure 7: Analysis of the mean absolute difference between observed batch-wise means $\hat{\mu}$ and stored BatchNorm means $\mu$ during testing for cPreResNet models trained with NS (Yu et al., 2018) or US (Yu & Huang, 2019). (a)-(b): The distribution of $|\mu - \hat{\mu}|$ across all layers. (c)-(d): The average value of $|\mu - \hat{\mu}|$ for each individual BatchNorm layer. (e)-(f): The correlation between the average of $|\mu - \hat{\mu}|$ and test set error.

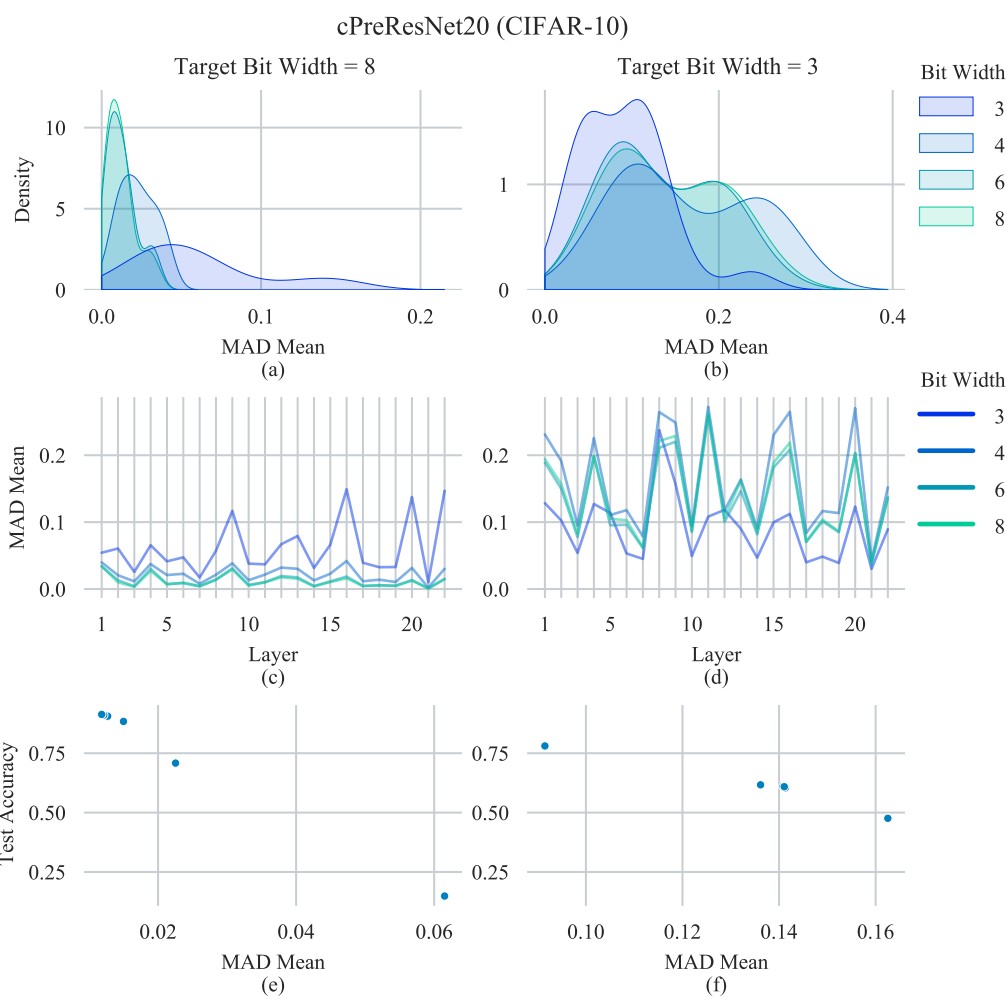

Figure 8: Analysis of the mean absolute difference between observed batch-wise means $\hat{\mu}$ and stored BatchNorm means $\mu$ during testing for cPreResNet models trained with different quantization bit widths. (a)-(b): The distribution of $|\mu - \hat{\mu}|$ across all layers. (c)-(d): The average value of $|\mu - \hat{\mu}|$ for each individual BatchNorm layer. (e)-(f): The correlation between the average of $|\mu - \hat{\mu}|$ and test set error.

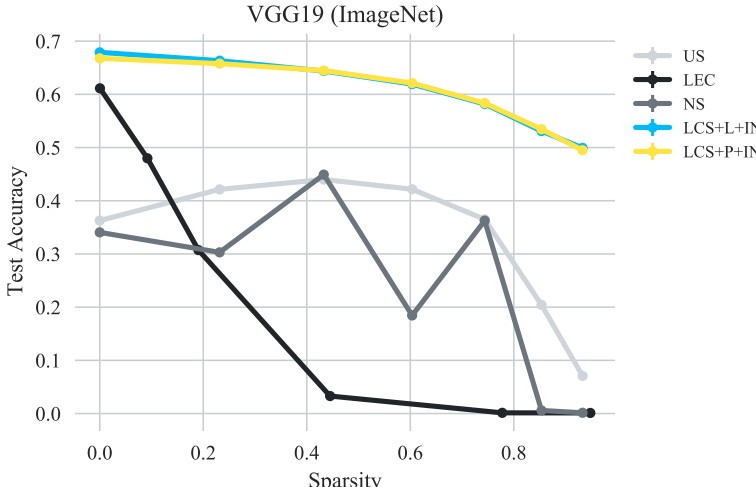

Figure 9: Our method for structured sparsity using a linear subspace (LCS+L+IN) and a point subspace (LCS+P+IN), compared to Universal Slimming (US) (Yu & Huang, 2019), Network Slimming (NS) (Yu et al., 2018), and Learning Efficient Convolutions (LEC) (Liu et al., 2017).

with $25\%$ probability, and $\boldsymbol{\alpha} = [U(\alpha_{\min}, \alpha_{\max})]$ with $50\%$ probability, where $U(a,b)$ samples uniformly in the range $[a, b]$. To warm up our sparsity rates, we choose

$$d = \max(1 - c/t, 0) \tag{2}$$
$$\gamma(\alpha) = (1 - \alpha)(1 - d), \tag{3}$$

where $c$ is the current iteration number, and $t$ is the total number of iterations in the first $80\%$ of training. At the beginning of training, $d = 1$, and $\gamma(\alpha) = 0$, corresponding to a sparsity level of $0$ for all values of $\alpha$. Once $80\%$ of training is finished, $d = 0$, and $\gamma(\alpha) = 1 - \alpha$ for the remainder of training. This corresponds to our final sparsity range.

## A.5 ADDITIONAL RESULTS

**Structured Sparsity:** We present results for VGG19 on ImageNet in the structured sparsity setting in Figure 9. Our method produces a better efficiency-accuracy trade-off than baselines. Note that VGG19 with GroupNorm (Wu & He, 2018) had a higher baseline accuracy than VGG19 with BatchNorm (Ioffe & Szegedy, 2015) (as noted in Section 4.3).

We also provide a table demonstrating the memory, flops, and runtime of structured sparsity models in Table 2.

**Unstructured Sparsity:** We present results for a wider range of sparsity values in our unstructured sparsity experiments in Figure 11. We change our sparsity limits from $[0.005, 0.05]$ (in Section 4.2) to $[0.025, 1.0]$. All other training details are unchanged. We find that our method is able to produce a higher accuracy over a wider range of sparsities than our baselines.

We also provide a table of unstructured sparsity results in the high sparsity regime (Table 3) and the wider sparsity regime (Table 4), showing memory usage and FLOPS.

**Quantization:** We present quantization results for ResNet18 in Figure 10. We find that our models approach the accuracy of models trained at a single bit width, and our models generalize better to other bit widths.

We also provide a table of quantization results in Table 5, showing memory usage of the models.

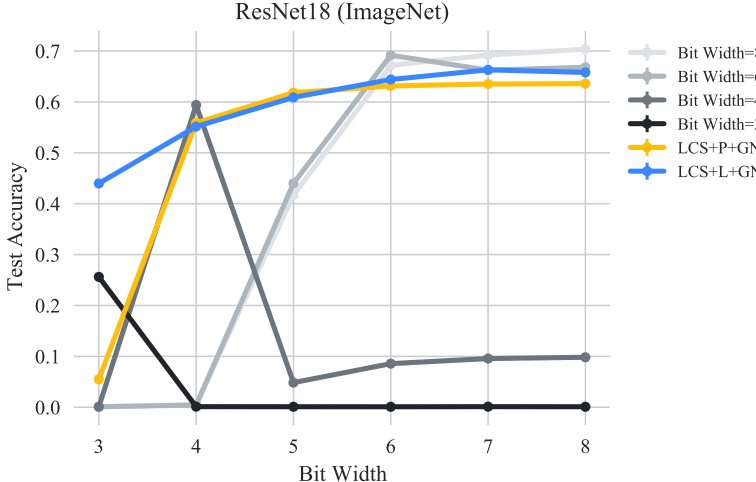

Figure 10: Our method for quantization using a linear subspace (LCS+L+GN) and a point subspace (LCS+P+GN) compared to networks trained for a particular bit width target.

Table 2: Results for structured sparsity. Note that models of a particular architecture and sparsity level all have the same runtime characteristics (memory, FLOPS, and runtime), so we only report one value. Runtime was measured on a MacBook Pro (16-inch, 2019) with a 2.6 GHz 6-Core Intel Core i7 processor and 16GB 2667 MHz DDR4 RAM. Memory consumption refers to the size of model weights in the currently executing model.

| | | | | | | | |
|---|---|---|---|---|---|---|---|
| cPreResNet20 (CIFAR-10) | Sparsity (%) | 0 | 43.491 | 60.614 | 74.655 | 85.614 | 93.491 |
| | FLOPS ($\times 10^6$) | 33.75 | 19.07 | 13.29 | 8.55 | 4.85 | 2.2 |
| | Memory (MB) | 0.87 | 0.49 | 0.34 | 0.22 | 0.12 | 0.06 |
| | Runtime (ms) | 3.13 | 2.64 | 2.09 | 1.83 | 1.64 | 1.28 |
| | Acc (LCS+P+IN) | 87.51 | **86.07** | **84.46** | **82.02** | **78.39** | **75.96** |
| | Acc (LCS+L+IN) | **88.32** | 85.69 | 83.69 | 80.84 | 77.5 | 75.31 |
| | Acc (US) | 70.62 | 83.13 | 81.11 | 62.18 | 40.04 | 21.81 |
| | Acc (NS) | 72.87 | 75.46 | 57.09 | 70.07 | 16.86 | 19.76 |
| ResNet18 (ImageNet) | Sparsity (%) | 0.0 | 42.91 | 59.89 | 73.88 | 84.89 | 92.91 |
| | FLOPS ($\times 10^6$) | 1814.1 | 1042.66 | 736.42 | 483.16 | 282.89 | 135.61 |
| | Memory (MB) | 46.72 | 26.67 | 18.74 | 12.2 | 7.06 | 3.31 |
| | Runtime (ms) | 45.85 | 30.34 | 22.51 | 14.31 | 9.84 | 6.02 |
| | Acc (LCS+P+IN) | 63.32 | 60.21 | **57.42** | **53.77** | **48.75** | **44.62** |
| | Acc (LCS+L+IN) | **64.21** | 59.3 | 56.18 | 52.73 | 48.35 | 44.5 |
| | Acc (US) | 58.91 | **60.39** | 53.76 | 44.72 | 22.51 | 8.34 |
| | Acc (NS) | 50.63 | 57.58 | 22.93 | 48.52 | 0.84 | 1.34 |
| VGG19 (ImageNet) | Sparsity (%) | 0.0 | 43.28 | 60.35 | 74.37 | 85.35 | 93.28 |
| | FLOPS ($\times 10^6$) | 19533.52 | 11008.56 | 7656.48 | 4911.33 | 2773.1 | 1241.81 |
| | Memory (MB) | 82.12 | 46.58 | 32.56 | 21.04 | 12.03 | 5.52 |
| | Runtime (ms) | 388.49 | 246.81 | 172.64 | 105.77 | 60.0 | 29.55 |
| | Acc (LCS+P+IN) | 66.77 | **64.47** | **62.11** | **58.35** | **53.45** | **49.5** |
| | Acc (LCS+L+IN) | **67.7** | 64.16 | 61.37 | 57.6 | 52.81 | 49.44 |
| | Acc (US) | 36.27 | 43.99 | 42.17 | 36.5 | 20.42 | 7.07 |
| | Acc (NS) | 34.05 | 44.91 | 18.44 | 36.26 | 0.57 | 0.14 |

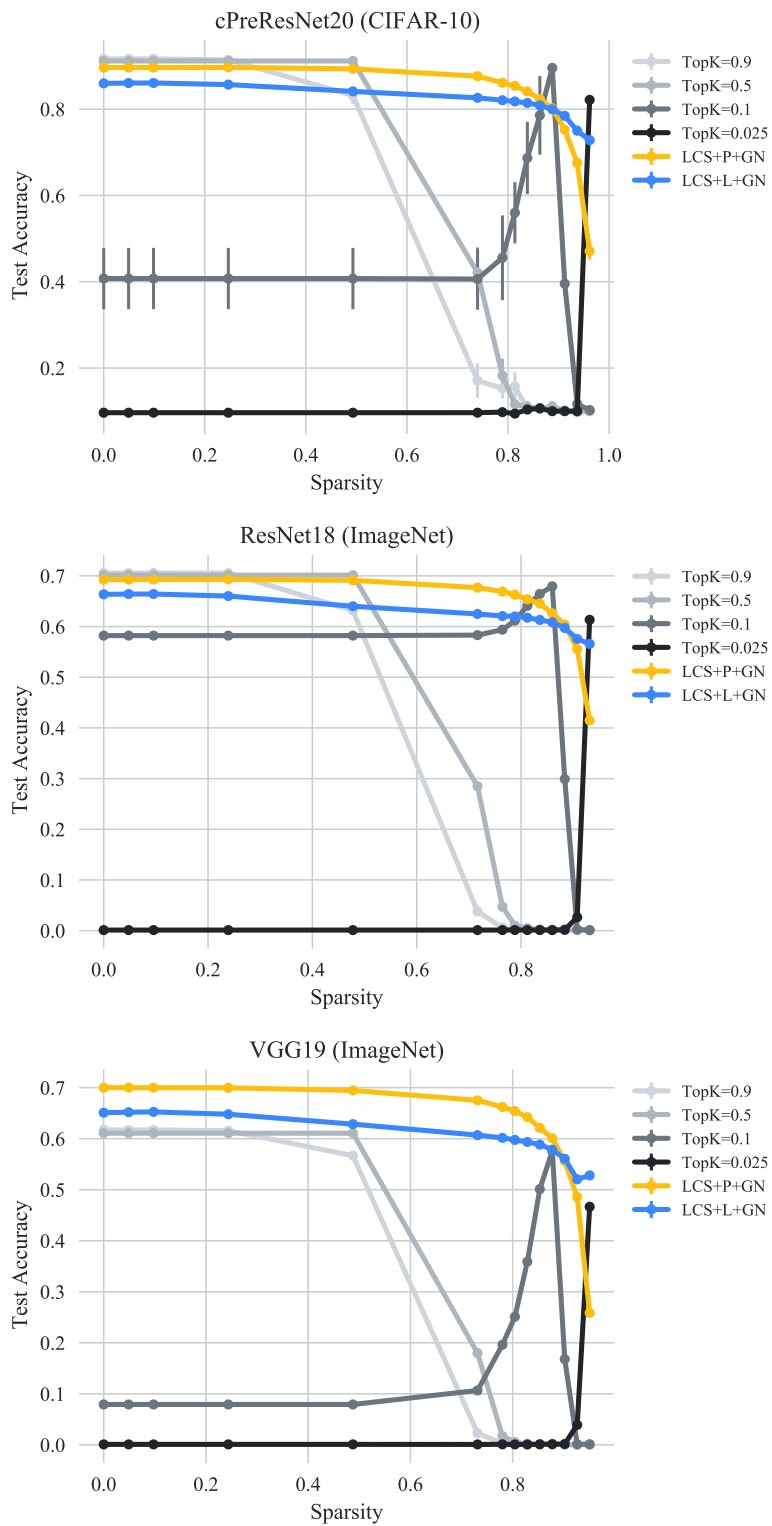

Figure 11: Our method for unstructured sparsity using a linear subspace (LCS+L+GN) and a point subspace (LCS+P+GN) compared to networks trained for a particular TopK target. The TopK target refers to the fraction of weights that remain unpruned during training.

Table 3: Results for unstructured sparsity in the high sparsity regime. Note that models of a particular architecture and sparsity level all have the same runtime characteristics (memory and FLOPS), so we only report one value. Runtime was not measured because it requires specialized hardware. So, we follow the standard practice of only reporting memory and flops. Memory consumption refers to the size of model weights in the currently executing model.

| | | | | | | | |
|---|---|---|---|---|---|---|---|
| cPreResNet20 (CIFAR-10) | Sparsity (%) | 95.66 | 96.15 | 96.64 | 97.14 | 97.63 | 98.12 |
| | FLOPS ($\times 10^6$) | 1.46 | 1.29 | 1.13 | 0.96 | 0.79 | 0.63 |
| | Memory (MB) | 0.04 | 0.03 | 0.03 | 0.02 | 0.02 | 0.02 |
| | Acc (LCS+P+GN) | 70.18 | 69.22 | 67.74 | **63.78** | 54.91 | 24.92 |
| | Acc (LCS+L+GN) | **73.2** | **69.5** | 64.42 | 54.77 | 42.62 | 30.61 |
| | Acc (TopK=0.04) | 14.83 | 12.83 | 10.8 | 9.66 | 10.01 | 9.79 |
| | Acc (TopK=0.02) | 10.25 | 10.53 | **78.7** | 10.56 | 10.05 | 10.09 |
| | Acc (TopK=0.01) | 10.02 | 9.97 | 9.96 | 10.64 | **59.2** | 10.79 |
| | Acc (TopK=0.005) | 10.0 | 10.08 | 9.81 | 10.03 | 10.0 | **41.44** |
| ResNet18 (ImageNet) | Sparsity (%) | 92.67 | 93.15 | 93.62 | 94.1 | 94.58 | 95.06 |
| | FLOPS ($\times 10^6$) | 169.42 | 160.94 | 152.46 | 143.98 | 135.51 | 127.03 |
| | Memory (MB) | 3.42 | 3.2 | 2.98 | 2.76 | 2.53 | 2.31 |
| | Acc (LCS+P+GN) | 51.5 | 51.1 | 50.37 | 48.62 | 44.8 | 30.69 |
| | Acc (LCS+L+GN) | **58.56** | **57.01** | 54.72 | **50.98** | 45.01 | 39.32 |
| | Acc (TopK=0.04) | 5.96 | 0.9 | 0.18 | 0.11 | 0.1 | 0.1 |
| | Acc (TopK=0.02) | 24.66 | 45.37 | **59.92** | 3.84 | 0.1 | 0.11 |
| | Acc (TopK=0.01) | 0.12 | 0.1 | 0.1 | 0.11 | **53.95** | 0.11 |
| | Acc (TopK=0.005) | 0.1 | 0.12 | 0.11 | 0.1 | 0.11 | **46.35** |

Table 4: Results for unstructured sparsity in the wide sparsity regime. Note that models of a particular architecture and sparsity level all have the same runtime characteristics (memory and FLOPS), so we only report one value. Runtime was not measured, because it requires specialized hardware (so most unstructured pruning works report memory and flops). Memory consumption refers to the size of model weights in the currently executing model.

| | | | | | | | |
|---|---|---|---|---|---|---|---|
| cPreResNet20 (CIFAR-10) | Sparsity (%) | 0.0 | 49.31 | 86.29 | 91.22 | 93.68 | 96.15 |
| | FLOPS ($\times 10^6$) | 33.75 | 17.11 | 4.62 | 2.96 | 2.13 | 1.29 |
| | Memory (MB) | 0.87 | 0.44 | 0.12 | 0.08 | 0.05 | 0.03 |
| | Acc (LCS+P+GN) | 89.64 | **89.34** | **82.34** | 75.24 | 67.55 | 47.1 |
| | Acc (LCS+L+GN) | 85.97 | 84.12 | 80.86 | **78.44** | **75.0** | 72.81 |
| | Acc (TopK=0.9) | **91.66** | 83.17 | 10.54 | 10.0 | 9.76 | 10.0 |
| | Acc (TopK=0.5) | 91.16 | 91.17 | 10.64 | 10.14 | 10.0 | 10.0 |
| | Acc (TopK=0.1) | 40.74 | 40.74 | 78.56 | 39.54 | 11.67 | 10.26 |
| | Acc (TopK=0.025) | 9.64 | 9.64 | 10.65 | 10.0 | 9.98 | **82.15** |
| ResNet18 (ImageNet) | Sparsity (%) | 0.0 | 47.77 | 83.59 | 88.37 | 90.76 | 93.15 |
| | FLOPS ($\times 10^6$) | 1814.1 | 966.32 | 330.49 | 245.72 | 203.33 | 160.94 |
| | Memory (MB) | 46.72 | 24.4 | 7.66 | 5.43 | 4.32 | 3.2 |
| | Acc (LCS+P+GN) | 69.25 | 69.12 | 64.53 | **60.36** | 55.58 | 41.48 |
| | Acc (LCS+L+GN) | 66.36 | 64.0 | 61.3 | 59.72 | **57.55** | 56.58 |
| | Acc (TopK=0.9) | **70.57** | 63.17 | 0.1 | 0.1 | 0.1 | 0.1 |
| | Acc (TopK=0.5) | 70.15 | **70.15** | 0.24 | 0.17 | 0.1 | 0.12 |
| | Acc (TopK=0.1) | 58.21 | 58.21 | **66.44** | 29.93 | 0.24 | 0.1 |
| | Acc (TopK=0.025) | 0.12 | 0.12 | 0.13 | 0.17 | 2.64 | **61.33** |
| VGG19 (ImageNet) | Sparsity (%) | 0.0 | 48.75 | 85.31 | 90.19 | 92.62 | 95.06 |
| | FLOPS ($\times 10^6$) | 19533.52 | 9822.65 | 2539.51 | 1568.42 | 1082.88 | 597.34 |
| | Memory (MB) | 82.12 | 42.09 | 12.06 | 8.06 | 6.06 | 4.06 |
| | Acc (LCS+P+GN) | **70.0** | **69.45** | **62.11** | 55.62 | 48.58 | 25.87 |
| | Acc (LCS+L+GN) | 65.1 | 62.85 | 58.85 | **56.07** | **52.04** | **52.8** |
| | Acc (TopK=0.9) | 61.72 | 56.67 | 0.1 | 0.0 | 0.0 | 0.1 |
| | Acc (TopK=0.5) | 61.07 | 61.07 | 0.15 | 0.09 | 0.06 | 0.1 |
| | Acc (TopK=0.1) | 7.91 | 7.91 | 50.13 | 16.8 | 0.13 | 0.09 |
| | Acc (TopK=0.025) | 0.09 | 0.09 | 0.1 | 0.16 | 3.91 | 46.66 |

Table 5: Results for quantization. Note that models of a particular architecture and quantization bit width all use the same memory, so we only report one value. Runtime was not measured, because it requires specialized hardware. Memory consumption refers to the size of model weights in the currently executing model.

| | Bit Widths | 8 | 7 | 6 | 5 | 4 | 3 |
|---|---|---|---|---|---|---|---|
| | Memory (MB) | 0.22 | 0.19 | 0.17 | 0.14 | 0.11 | 0.08 |
| | Acc (LCS+P+GN) | 89.97 | 90.0 | 89.88 | **89.26** | 86.25 | 65.26 |
| cPreResNet20 | Acc (LCS+L+GN) | 87.44 | 88.09 | 87.9 | 86.93 | 84.2 | **75.16** |
| (CIFAR-10) | Acc (Bit Width=8) | **91.36** | **91.02** | 90.47 | 87.98 | 65.91 | 16.65 |
| | Acc (Bit Width=6) | 91.07 | 90.89 | **91.26** | 87.12 | 63.1 | 18.78 |
| | Acc (Bit Width=4) | 84.93 | 84.65 | 84.77 | 82.37 | **88.22** | 25.19 |
| | Acc (Bit Width=3) | 55.71 | 55.56 | 57.01 | 55.66 | 44.83 | 73.89 |
| | Memory (MB) | 11.69 | 10.23 | 8.77 | 7.31 | 5.84 | 4.38 |
| ResNet18 | Acc (LCS+P+GN) | 63.59 | 63.51 | 63.15 | **61.82** | 55.89 | 5.48 |
| (ImageNet) | Acc (LCS+L+GN) | 65.79 | 66.31 | 64.4 | 60.89 | 55.16 | **43.98** |
| | Acc (Bit Width=8) | **70.36** | **69.2** | 67.18 | 41.67 | 0.54 | 0.08 |
| | Acc (Bit Width=6) | 66.8 | 66.26 | **69.17** | 44.0 | 0.43 | 0.1 |
| | Acc (Bit Width=4) | 9.82 | 9.57 | 8.57 | 4.84 | **59.39** | 0.1 |
| | Acc (Bit Width=3) | 0.1 | 0.12 | 0.09 | 0.1 | 0.12 | 25.61 |
| | Memory (MB) | 20.542 | 17.974 | 15.406 | 12.839 | 10.271 | 7.703 |
| VGG19 | Acc (LCS+P+GN) | 57.83 | 57.74 | 57.24 | 55.64 | 47.11 | 0.25 |
| (ImageNet) | Acc (LCS+L+GN) | **63.72** | **64.27** | **62.08** | **57.87** | 50.05 | **33.46** |
| | Acc (Bit Width=8) | 60.85 | 59.32 | 56.13 | 39.73 | 0.3 | 0.09 |
| | Acc (Bit Width=6) | 56.52 | 55.35 | 59.89 | 23.24 | 0.16 | 0.1 |
| | Acc (Bit Width=4) | 0.13 | 0.13 | 0.14 | 0.11 | **51.33** | 0.1 |
| | Acc (Bit Width=3) | 0.12 | 0.1 | 0.1 | 0.12 | 0.09 | 20.72 |

