# OpenReview forum: "LCS: Learning Compressible Subspaces for Adaptive Network Compression at Inference Time"
_ICLR.cc/2022/Conference — ICLR 2022 Submitted_

### Official Review · Reviewer_EG95 · 2021-11-02

**Correctness:** 3
**Technical Novelty And Significance:** 3
**Empirical Novelty And Significance:** 3
**Recommendation:** 8
**Confidence:** 4

**Main Review:**

The paper is well written and easy to follow. The ideas of constructing a linear subspace and using the function $f(w(\alpha), \gamma (\alpha))$ to perform compression during inference are novel. The analysis of BN parameters in adjustment provides quantitative analysis in this area. The experiments are exhaustive and can well support their ideas.

However, as the paper claims, they bias the subspace to contain high-accuracy solutions at one end and high-efficiency solutions at the other end. In my understanding, two endpoints are using the same network architecture. How to train a network to obtain $w_1$ and $w_2$ in this case?

**Summary Of The Paper:**

The paper proposes to learn compressive subspaces which can adaptively compress the network during inference. It constructs either linear subspace or a single endpoint for compression. It replaces BN with GroupNorm to avoid re-calibrating during inference or after adjustment. Their method is evaluated in three different scenarios: structured sparsity, unstructured sparsity, and quantization.

**Summary Of The Review:**

I think the paper is well written; the method is novel and interesting; the experiment can well support the claims. Just need to clarify some details

---

> ### Author Response · Authors · 2021-11-16
> **Rebuttal Response**
>
> Thank you for the feedback. We respond to your question below:
>
> > However, as the paper claims, they bias the subspace to contain high-accuracy solutions at one end and high-efficiency solutions at the other end. In my understanding, two endpoints are using the same network architecture. How to train a network to obtain w1 and w2 in this case?
>
> We aren't sure we understand the question. We have a single network architecture, and two sets of weights w1 and w2 which define endpoints of the subspace (both sets of weights correspond to the same network architecture). During training, we sample a point $w^*$ along the line segment joining w1 and w2 (hence $w^*$ has the same architecture as w1 and w2), and compute the loss using this set of weights which then allows us to perform gradient descent updates on w1 and w2. This procedure is explained in section 3.1. If this doesn’t answer your question, can you please clarify your question?

---

### Official Review · Reviewer_jBcH · 2021-11-03

**Correctness:** 3
**Technical Novelty And Significance:** 1
**Empirical Novelty And Significance:** 2
**Recommendation:** 3
**Confidence:** 4

**Main Review:**

The novelty of this paper looks limited. It is an extension of recent work on learning network subspaces. Meanwhile, compared to existing works with adaptive networks, the advantages of this work are the finer-grained compress level and no need to recalibrate BN. The benefit of such improvements looks trivial.

Many vital details are missing in this paper. The detailed form of the compression level function $\gamma(\alpha)$ and the compression function $f(\omega, \gamma)$ are not provided. These are the core of the algorithm and the authors need to show them.

Also, the authors do not explain what determines the dimension n of the stochastic function $\alpha$. If $\alpha$ controls the sparsity at the level of each weight, n will be extremely large, and the training overhead of the proposed algorithm is extremely large because they need to do n forward passes and backward passes for each batch. Even $\alpha$ controls the sparsity at the layer level, the training overhead will still be formidable. I cannot find much information about n in this paper. The authors need to provide more details about the dimension n of $\alpha$.

**Summary Of The Paper:**

This paper proposes to balance the inference accuracy and efficiency by training a subspace of neural networks and then adapting the network within the subspace at inference time.

**Summary Of The Review:**

Many important details are missing in this paper. For example, the finer-grained compression level is a major selling point of this paper, but the authors did not even provide the compression level function $\gamma(\alpha)$.

---

> ### Author Response · Authors · 2021-11-16
> **Rebuttal Response Part 1**
>
> Thank you for the feedback. We respond per paragraph:
>
> > The novelty of this paper looks limited...
>
> 1. Our exploration of subspaces through the lens of model compression is novel, and has interesting implications for our understanding of the loss landscape. Garipov et al (https://arxiv.org/pdf/1802.10026.pdf) showed that there exist simple paths in weight space that connect optima along which performance remains relatively constant. Wortsman et al (https://arxiv.org/pdf/2102.10472.pdf) showed that we can find simplexes in weight space along which performance remains relatively constant. Both of these works engender a better understanding of the geometry of neural network's loss landscape. Our work extends this understanding by showing that we can find regions in weight space that robust to sparsity/quantization operations, providing a smooth tradeoff in terms of accuracy/efficiency. This is an exciting discovery.
>
> 2. Providing fine-grained compression levels and avoiding BatchNorm recalibration are critical for flexibly adjusting model size on-device to ever-changing resource constraints (see section 1 for motivation). Here are a few key points:
> - We want to operate at arbitrarily fine granularities to maximize resource usage. If only a few settings are given, a system using US/NS with width factors [0.25, 0.50, 0.75, 1.0] might be forced to use a model with width factor 0.25 even if a width factor 0.49 could have been run. This results in degraded accuracy.
> - Consider the cost of storing every possible width factor for US/NS between 0.25 and 1.0 for ResNet18 (which has 11.8M parameters, and 7808 BatchNorm parameters). Since the largest feature map is size 512, the cost is 512*0.75*7808=3.0M, which is a 25% overhead. That overhead is very significant on edge devices (especially when running mulitple models, etc. Every bit of storage counts on FPGAs, ASICs, etc.). Even if the granularity is reduced, the overhead will still be significant on edge compute devices (and the problem above will arise). By contrast, our LCS+P method has no overhead. Note that the situation is more dire in unstructured sparsity, in which much finer granularities are achievable.
> - Removing recalibration enables more potential scenarios. Consider a multi-headed model in which compute in the shared base, and all the heads, needs to vary dynamically (e.g. different sparsities are chosen for the heads and the base). If recalibration is required, we would need to deploy a combinatorial number of BatchNorm parameters (for each possible size setting for the base and heads). This isn't feasible.
> - Our method also opens the opportunity for exploring compressions that focus on tuning individual layers separately (e.g. pruning a single layer that has particularly high memory usage, in the case where memory is restricted but compute is not). Supporting this with recalibration requires deploying a combinatorial set of BatchNorm parameters.
> - In light of the above two points, our work builds towards future works leveraging the advantages of deployment that doesn't require recalibration.
>
> > Many vital details are missing in this paper. The detailed form of the compression level function γ(α) and the compression function f(ω,γ) are not provided. These are the core of the algorithm and the authors need to show them.
>
> Please see the last paragraph of section 3 (right before the start of section 4). It explains that gamma and f are dependent on the compression method used, and that detailed descriptions are in Section 4.1 (structured pruning), 4.2 (unstructured pruning), and 4.3 (quantization).

---

> ### Author Response · Authors · 2021-11-16
> **Rebuttal Response Part 2**
>
> > Also, the authors do not explain what determines the dimension n of the stochastic function α.
>
> The role of 'n' is explained in Section 3.4. It controls the number of forward passes before a backward pass is used. It is an optimization hyperparameter introduced in the US baseline. The values of 'n' are implicitly given in 4.1, 4.2, and 4.3, when the stochastic function is described.
>
> $\alpha$ controls the position in our subspace, and it is always a scalar. It controls the sparsity level gamma, which is a function $\gamma(\alpha)$. Note that in the unstructured sparsity case, $\gamma(\alpha)$ gives the fraction of pruned weights - it is a scalar. There isn't an $\alpha$ value for each weight or for each layer, there's only a single $\alpha$ value.
>
> Note that our notation uses three related symbols: (1) $\alpha$ (not bolded) is a scalar which controls the position in the neural network subspace, (2) $\boldsymbol{\alpha_n}$ is a stochastic function which samples a vector of size 'n' (I will describe 'n' in a moment), and (3) $\boldsymbol{\alpha}$ is a vector of size 'n' (e.g. an output from calling the stochastic function $\boldsymbol{\alpha_n}$).
>
> The purpose of $\boldsymbol{\alpha_n}$ is to generate a set of values $\boldsymbol{\alpha}$, which will be used during training. Take a look at Algorithm 1. A for-loop traverses over all $\alpha$ values in $\boldsymbol{\alpha}$. It computes the loss function, then continues to the next alpha value.
>
> In other words, 'n' is a hyperparameter that tells you "how many different $\alpha$ values are chosen for a forward pass, before gradient updates are applied". In all structured sparsity experiments, we use "n=4", since this is what baselines NS and US use (see the discussion in Section 4.1). For all other experiments, we use "n=1", because it would be unfair to use extra training iterations for our method compared to baselines. We demonstrate the "n" value implicitly in sections 4.1, 4.2, and 4.3 when we describe the stochastic function (note that, $\boldsymbol{\alpha}$ is a vector of length 4 in section 4.1, and it is a 1-element vector in section 4.2 and 4.3).

---

### Official Review · Reviewer_FDfN · 2021-11-03

**Correctness:** 2
**Technical Novelty And Significance:** 2
**Empirical Novelty And Significance:** 1
**Recommendation:** 3
**Confidence:** 4

**Main Review:**

Strengths: the paper is well motivated, as adaption to runtime available resource is important.

Weaknesses:
* The method is a direct extension of the learning subspace method.
* There are important details missing from the paper. E.g., there is no info on how to generate a alpha via the state of hardware in runtime. This creates severe difficulty in understanding and reproducing the method.
* There is no material to measure hardware performance. For example, only the accuracy of the classification models are given, but memory bandwidth, latency or FPS are not available to quantitatively measure the advantage.
* The paper is not properly peer-compared. For example, the work is not well compared with pruning and quantification methods.

Questions for the Author(s):
* please elaborate on the definition of the compression function f and the intuition behind?
* how to choose the hyper-parameter alpha in a hardware run-time?
* What will happen to the arch of a model, if pruning is also performed?
* If using this method in a hardware, how to change the quantization meta-parameters(scale and zero-point) accordingly?



**Summary Of The Paper:**

The paper present a method for learning a compressible subspace of neural networks that contains a fine-grained spectrum of models that range from highly efficient to highly accurate. The proposed method allows choosing the proper point of trade-off between accuracy and efficiency at inference time, according to the available resources. There are also efforts to reduce the runtime tweaking overhead like replacing BatchNorm with GroupNorm.

**Summary Of The Review:**

The paper proposes a method that reasonably extends the learning subspace method to allow performing accuracy-efficiency tradeoff according to runtime available resource. The method has been evaluated on several classification tasks and find to be useful.

However, the paper does not clearly explain how the compression is performed, with important details like choice of alpha missing. The measurement of speedup is not that quantitative, lacking realworld test stats. It is very difficult to evaluate the contribution of this paper under these conditions.

---

> ### Author Response · Authors · 2021-11-16
> **Rebuttal Response Part 1**
>
> Thank you for the feedback. We respond per-paragraph:
>
> > The method is a direct extension of the learning subspace method.
>
> Sorry, we do not understand the objection - many excellent works are extensions of existing works. Can you clarify the issue?
>
> We assume your objection is regarding novelty and/or the significance of our contribution. Please see the end of Section 1 for a summary of our novel contributions. To facilitate discussion, we summarize here as well:
> - we develop a method for producing a variety of efficiency-accuracy trade-offs, allowing on-device model compression in real time, without specifying the compression level beforehand (this has never been done before).
> - we demonstrate an intriguing new use for network subspaces - one can find a linear region in weight space with low loss and a variety of efficiency-accuracy trade-offs. This builds on work by Garipov (https://arxiv.org/pdf/1802.10026.pdf) and Worstman (https://arxiv.org/pdf/2102.10472.pdf) and deepens our understanding of the loss landscape.
> - our method requires no BN recalibration for deployment of models, unlike previous methods (US/NS). This reduces the number of deployed parameters (which is very significant in edge computing on FPGA, ASIC, etc.). Our LCS+P method has no parameter overhead, whereas deploying many sets of BN parameters to support dynamic on-device compression can be costly.
> - We provide arbitrarily fine-grained efficiency-accuracy trade-offs, which previous works have not investigated.
> - We evaluate our method in three different compression settings, showing the flexibility of our method. Previous works investigate only a single setting.
>
> > There are important details missing, e.g. no info on how to generate an alpha via the state of hardware runtime.
>
> Generating alpha (to choose the compression level) is the responsibility of the user. Generally speaking, compression papers expect the users to decide their desired compression level based on their application and target device. Our situation is similar - the user will choose an alpha value corresponding to the desired compression level. The difference between our method and standard compression is that the user has the flexibility to change the compression level at any time after deployment, but the user must still decide the compression level (perhaps by querying the operating system to understand overall process load). Note that this assumption is also made by our NS and US baselines, which assume that the user chooses an appropriately sized network.
>
> > There is no material to measure hardware performance. For example, only the accuracy of the classification models are given, but memory bandwidth, latency or FPS are not available to quantitatively measure the advantage.
>
> We will add a table showing memory and compute to the Appendix. (Section A.5). The sparsity and quantization directly determine the memory and runtime of the models. Note that all models for a given sparsity method and sparsity rate will have the same runtime and memory bandwidth, so the comparison in Section 4 is fair.
>
> > The paper is not properly peer-compared. For example, the work is not well compared with pruning and quantification methods.
>
> Our work should not be compared to standard pruning or quantization techniques. They do not allow for dynamic inference-time compression. Instead, we should compare with other methods for dynamic inference-time compression (such as US and NS). Note that previous works such as US and NS only provide results for a single domain - structured sparsity. We provide results for three domains - structured sparsity, unstructured sparsity, and quantization. Therefore, we believe our evaluation is extensive.
>
> > Please elaborate on the definition of the compression function f and the intuition behind?
>
> Please see Section 3.1 for the introduction of f, and section 4.1, 4.2, and 4.3 for the precise formulation of f in each experimental scenario. Briefly, the compression function f represents the compression method that we use. Thus, f corresponds to either "structured pruning" (section 4.1), "unstructured pruning" (section 4.2), or "quantization" (section 4.3). The function f simply represents "compression methods (such as pruning and quantization) from the literature". The compression function takes in network weights and a compression level, and produces a compressed model. For instance, in the case of structured sparsity, f takes in the original network weights, and a value representing the fraction of channels to delete. The output of f is a version of the network with those channels deleted. In the case of quantization, f takes in the original network, and the desired bit-width, and outputs a quantized network at the desired bit-width.
>
> > How to choose the hyper-parameter alpha in a hardware run-time?
>
> See above comment.

---

> ### Author Response · Authors · 2021-11-16
> **Rebuttal Response Part 2**
>
> > What will happen to the arch of a model, if pruning is also performed?
>
> Sorry, we are not sure we understand your question. For structured sparsity, we apply standard channel pruning (as used in US and NS) and for unstructured sparsity we apply TopK pruning to our architecture to produce a compressed model. Both practices appears in previous literature. The architecture structure remains unchanged as a result of these operations (of course, structured pruning deletes some input and output channels, but the overall architecture is the same). Does this answer your question?
>
> > If using this method in a hardware, how to change the quantization meta-parameters(scale and zero-point) accordingly?
>
> The same scale and zero-point are used at all bit widths.

---

### Official Review · Reviewer_Dpge · 2021-11-05

**Correctness:** 3
**Technical Novelty And Significance:** 3
**Empirical Novelty And Significance:** 3
**Recommendation:** 3
**Confidence:** 4

**Main Review:**

=== Strengths ===

Table 1 is useful in summarizing the related works and the claimed advantages for LCS.

The experiment section listed a significant amount of detail on hyperparams and methods.

The small discussion and figures on batch norm stats shifts were interesting and helpful for motivating group and instance norm for this application.

This area of adaptive inference is becoming more and more important with larger models and specialized big-little architectures.

=== Weaknesses ===

It is unclear how these networks switch between models at inference time, which of course should depend on the type of compression used.  For sparsity, this seems like it would require dynamically pruning the model at inference time which seems very dangerous. For unstructured sparsity, this may require special hardware for taking advantage of that unstructured sparsity. For quantization, this may require hardware can support fine-grained switching of the quantization bitwidth.

My understanding is that batch norm stats have to be recomputed for NS and US in a post-training way but not necessarily at inference time. It doesn't seem fair to avoid this step since it can be done before model deployment and takes a fraction of the training time. Leaving it out in the evaluation also nullifies the comparison if this is not fair.

The claim that other methods need additional batch norm params and cannot support fine-grained compression level is mostly correct, but the importance of this seems overstated. In practice, it seems more reasonable to chose a smaller subset of model configurations that can be fully tested before deployment, and the batch norm params should be nearly negligible compared to the weights. Also, quantization LCS does of course limit the number of compression levels, and structured pruning LCS limits the number of compression levels to the number of channels (which is similar to US).

The existence of gamma and alpha together is confusing to follow. Since the parameterization in linear, it seems like only one of these should be necessary.

There should be other works included for building robust compressible models, e.g. Robust Quantization (Neurips20).

The writing is clear but repetitive in some areas. For example, I believe there are 5 sentences talking about being inspired by Wortsman in the first few pages.

=== Questions ===
In the Related Works, the neural subspace method is described as operating on simplices, but the description in Section 3.1 seems to be on lines. Is this deliberate?

Please correct me if I'm wrong but isn't the unstructured compressible point method Dropout? There might be differently weighted probabilities and dynamic dropout probabilities, but they seem fundamentally the same.

For quantization, what hardware supports dynamic fine-grained switching from 3-8 bits? How are the pruned channels or pruned individual weights chosen at runtime in an adaptive way?

**Summary Of The Paper:**

This paper introduces learnable compressible subspaces, which attempts to learn a set of models that can be switched at inference time to adapt to different resource requirements. This work is motivated by previous work in neural subspaces and slimmable networks. It is evaluated on CIFAR10 and ImageNet and compared against other recent works for adaptable inference models. These results show under certain conditions LCS can maintain higher accuracies at larger sparsities compared to other works.

**Summary Of The Review:**

This paper is an interesting proposal that attempts to apply the ideas of neural subspaces to produce a set of compressed models at varying points on the accuracy / efficiency curve. Yet, these methods in the end seem more about learning robust compressible models and stray far from the original neural subspace idea, especially with compressible points. In my current understanding, these networks seem to have no demonstrated advantage to universally slimmable networks, which have a simpler validation process, runtime switching method, and more intuitive training procedure. The comparison against these networks and others needs to be better justified since I currently do not understand why fine-tuning is not allowed. If I misunderstood the method significantly, I would be willing to increase my score, but currently I suggest rejecting the paper.

---

> ### Author Response · Authors · 2021-11-16
> **Rebuttal Response (Part 1)**
>
> Thank you for the feedback. We respond per paragraph:
>
> > It is unclear how these networks switch between models at inference time...
>
> 1. Our method discusses on-the-fly model compression in Section 3, which is used to generate a compressed model (and we justify its real-time nature in Appendix A.1). The same procedure is used for training and inference - an alpha value is chosen (which corresponds to a compression level), and the compressed network is computed. Note that the selection of compression level (determined by alpha) is not discussed, because this is the responsibility of the user (just like how NS and US assume that the user choose the compression level).
>
> 2. Our results indicate that accuracy is maintained at varying compression levels with dynamic pruning. Please clarify what aspecs of dynamic pruning you believe are dangerous.
>
> 3. Hardware support for unstructured sparsity or quantization is in a nascent stage, and is still evolving. This issue is not unique to our paper, it applies to every unstructured sparsity paper and every low-bit quantization paper that has been published. Both are rich areas of research that have practical value in motivating hardware design. We believe that current lack of hardware support does not diminish the value of exploratory algorithmic development. We believe our exploration is valuable in understanding run-time accuracy-effiency trade-offs even though hardware development is ongoing.
>
> > My understanding is that batch norm stats have to be recomputed...
>
> Our goal is to produce a fine-grained efficiency-accuracy trade-off. In this case, we cannot recompute BatchNorm statistics before deployment without potentially significant on-device overhead (see related comment below for details, and scenarios in which BatchNorm statistics cannot be recomputed).
>
> > The claim that other methods need additional batch norm params ... is mostly correct, but...
>
> 1. Storing copies of BatchNorm statistics can be used in scenarios in which plenty of resources are available. But our focus is on real-world low-compute settings, in which problems arise with this approach:
> - We want to operate at arbitrarily fine granularities to maximize usage of available resources. If only a few BatchNorms are stored, then a system using US/NS with width factors [0.25, 0.50, 0.75, 1.0] might be forced to use a model with width factor 0.25 even if a width factor 0.49 could have been run. This results in degraded accuracy.
> - To remedy this, one could store all possible BatchNorm statistics. Consider the cost of storing statistics for every possible width factor for US/NS between 0.25 and 1.0 for ResNet18 (which has 11.8M parameters, and 7808 BatchNorm parameters). Since the largest feature map is size 512, the cost is 512 $\times$ 0.75 $\times$ 7808=3.0M, which is a 25% overhead. That overhead is very significant on edge devices (especially when running multiple models. And, every bit of storage counts on FPGAs, ASICs, etc.). Even if the granularity is reduced, the overhead will still be significant on edge compute devices (and the previous issue of granularity may arise). By contrast, our LCS+P method has no overhead. Note that the situation is more dire in unstructured sparsity, in which much finer granularities are achievable.
> - Removing recalibration enables more potential scenarios. Consider a multi-headed model in which compute in the shared base, and all the heads, needs to vary dynamically (e.g. different sparsities are chosen for the heads and the base). If recalibration is required, we would need to deploy a combinatorial number of BatchNorm parameters (for each possible size setting for the base and heads). This isn't feasible.
> - Our method also creates the opportunity for exploring compressions that focus on tuning individual layers separately (e.g. pruning a single layer that has particularly high memory usage, in the case where memory is restricted but compute is not). Supporting this with recalibration requires deploying a combinatorial set of BatchNorm parameters (e.g. a set for each combination of layer sizes).
>
> 2. Indeed, quantization limits the granularity of trade-offs. We still provide this investigation to show the generality of our method, which we believe to be enlightening in understanding the performance of network subspaces in a previously-uninvestigated scenario. We agree that in this use case, recalibration may suffice, but neither NS nor US investigate this domain. Their exploration is limited to structured sparsity. Though structured sparsity is also technically limited in granularity, it provides a much finer-grained trade-off, and recalibrating for every trade-off imposes significant overhead (see (1) above).

---

> > ### Comment · Reviewer_Dpge · 2021-12-01
> > **Response**
> >
> > I appreciate the detailed rebuttal. You provided real network examples, clarifications, and corrections (I briefly forgot that a line was a simplex).
> >
> > A main concern is still the practicality of these networks, and the specifics of how they can switch compression levels at run time. I understand how a given model can be sampled post-training, e.g. in the pruning variant: select an alpha, get the compression level, then apply that degree of sparsity to the network. Yet, how does this work dynamically when the battery is running low on mobile device? This case was mentioned in the abstract. Do you select a different alpha (corresponding to more compression), then dynamically prune the model on the edge device to a higher sparsity?
> >
> > You are right to point out that this paper cannot be judged for the early stages of hardware support for unstructured networks.

---

> > > ### Author Response · Authors · 2021-12-01
> > > **Regarding Details of Switching Compression Levels at Runtime**
> > >
> > > Thank you for your reply, and thank you again for the thoughtful review.
> > >
> > > Your understanding of the scenario is correct. Suppose battery is running low, and we wish to compress the on-device model using structured sparsity. We select an alpha corresponding to the desired compression level, then calculate the compressed model using our compression function ‘f’. It’s up to the user to determine the desired compression level based on the amount of available battery (and the compression level is used to determine alpha).
> > >
> > > Note that some of the details of how the compressed model is calculated and stored are dependent on the compression method. For example, in the structured sparsity case, it is not necessary to store the compressed model weights separately (since they are a subset of the full model weights). We can instead just use the original model weights, but ignore the weights corresponding to channels that we are no longer using when we compute a forward pass. In this way, we incur no storage overhead. We can just set alpha, and when performing our forward pass, the inference engine implementation would know how many channels to ignore based on the setting of alpha. Note that this optimization is trivial in PyTorch: you would just need to (1) take a slice of your convolutional kernel using the proper indices as determined by alpha, then (2) call your convolution function as normal. The implementation should be similarly easy in an on-device ML framework. (Note that US and NS would also require this implementation detail to avoid storing extra weights, so our method doesn’t introduce extra complication.)
> > >
> > > In the unstructured sparsity case, we can do something similar - we can apply sparsity during the forward pass by ignoring values in the weight tensor that are below a given threshold. (The threshold needs to be recalculated when alpha changes, but can be stored afterwards until alpha changes. See Appendix Section A.1 for a justification of the threshold calculation being real-time). The precise details of how this works would depend on the implementation of the sparse convolution as well as the nature of the hardware support for sparsity. Note that our baselines also require this feature.

---

> ### Author Response · Authors · 2021-11-16
> **Rebuttal Response (Part 2)**
>
> > The existence of gamma and alpha together is confusing to follow...
>
> Conceptually, alpha determines where in weight-space your model lies. Gamma determines the compression ratio. We felt it important to introduce two variables to keep these concepts clear. It's true that gamma is fully determined by alpha (thus we write it as a function of alpha in our notation), but since the precise formula depends on the compression scenario, we felt it was important to include both.
>
> > There should be other works included (Robust Quantization)
>
> Thank you. We will cite this work (https://arxiv.org/pdf/2002.07686.pdf), but we will not add their results to our graphs, because they do not provide results that are directly comparable to ours:
> - From reading their open-source code (https://github.com/moranshkolnik/RobustQuantization), it looks like the activation quantization occurs before (not after) skip connections. We put quantization after skip connections to ensure the input to the following convolution is quantized. This difference in compression scheme can cause accuracy discrepancies.
> - Their method appears to use distillation, which we do not. This makes the numbers that they reported in Figure 5 incomparable to ours, since we trained without distillation. (We could train with distillation in principle, but we chose not to for simplicity.
> - In their quantization-aware training results (Figure 5), they leave the activation bit widths fixed for almost all of their data points, so results are not comparable to ours (since we keep activation bit widths equal to weight bit widths). Their post-training quantization results (Table 1) appear to require an optimization (LAPQ, https://arxiv.org/pdf/1911.07190.pdf) that requires data and cannot be run on low-compute devices.
>
> > Writing is repetitive
>
> Thank you for the feedback, we will make edits accordingly.
>
> > In the Related Works, the neural subspace method is described as operating on simplices, but the description in Section 3.1 seems to be on lines. Is this deliberate?
>
> Yes, this is deliberate. Lines are a special case of the subspace method. We focus on lines for clarity since we use it for experimentation.
>
> > Is unstructured compressible point method the same as Dropout?
>
> Our compressible point method differs from dropout in several ways. The deletion of weights is deterministic (based on the weight magnitude) rather than probabilistic. The expected value of the number of weights removed in an iteration of Dropout is fixed (e.g. with dropout probability p=0.5, we will remove about half the weights on each iteration). By contrast, our method varies the number of weights removed at each iteration. Models trained with dropout are generally evaluated without dropout, whereas our method specifically prescribes evaluation with sparsity.
>
> There are conceptual similarities between Dropout and TopK pruning. But it's much more accurate to say that "our LCS+P method for unstructured sparsity is TopK pruning, with TopK values sampled at every iteration" than it is to say "our LCS+P method is dropout".
>
> > For quantization, what hardware supports dynamic fine-grained switching from 3-8 bits? How are the pruned channels or pruned individual weights chosen at runtime in an adaptive way?
>
> See our comment in Part 1 of our response, regarding hardware and pruning levels.

---

### Public Comment · ~Gregory_Benton1 · 2021-11-13
**Missing proper citation and discussion of related work on subspace learning**

We enjoyed reading your paper; however, we request you revisit the Neural Network Subspaces section of your related work and the discussion of learning subspaces in the introduction. Our work _Loss Surface Simplexes for Mode Connecting Volumes and Fast Ensembling_ (https://arxiv.org/abs/2102.13042) appeared at ICML 2021 along with _Learning Neural Network Subspaces_ (https://arxiv.org/abs/2102.10472) and independently introduced the idea of subspace learning in neural networks through the definition of low loss simplexes in parameter space.

In a discussion of related work regarding learning subspace simplexes, our work is highly relevant and should be cited and discussed accordingly.

---

> ### Author Response · Authors · 2021-11-16
> **Added Your Work**
>
> Thank you, we have added your work to our related works section.

---

### Author Response · Authors · 2021-11-16
**Summary of Paper Update**

- Minor adjustments in to text (as per reviewer feedback).
- Added related neural network subspace work.
- Added an experimental detail to the appendix in the structured sparsity case (Section A.3).

---

### Decision · Program_Chairs · 2022-01-20

**Decision:**

Reject

**Comment:**

This paper proposed a method for adaptive network compression at inference time. However, the paper contains various issues raised by the reviewers that needs to be addressed.